# Role of the Ubiquitin Proteasome System (UPS) in the HIV-1 Life Cycle

**DOI:** 10.3390/ijms20122984

**Published:** 2019-06-19

**Authors:** Vivian K. Rojas, In-Woo Park

**Affiliations:** Department of Microbiology, Immunology, and Genetics, University of North Texas, Health Science Center, Fort Worth, TX 76107, USA; vivian.rojas@unthsc.edu

**Keywords:** HIV-1, ubiquitin, ubiquitin proteasome system, HIV-1 viral proteins, ubiquitin, E3 ligase, proteasomal degradation

## Abstract

Given that the ubiquitin proteasome system (UPS) is the major protein degradation process in the regulation of a wide variety of cellular processes in eukaryotic cells, including alteration of cellular location, modulation of protein activity, and regulation of protein interaction, it is reasonable to suggest that the infecting HIV-1 and the invaded hosts exploit the UPS in a contest for survival and proliferation. However, to date, regulation of the HIV-1 life cycle has been mainly explained by the stage-specific expression of HIV-1 viral genes, not by elimination processes of the synthesized proteins after completion of their duties in the infected cells, which is also quintessential for understanding the molecular processes of the virus life cycle and thereby HIV-1 pathogenesis. In fact, several previous publications have indicated that the UPS plays a critical role in the regulation of the proteasomal degradation of viral and cellular counterparts at every step of the HIV-1 life cycle, from the virus entry to release of the assembled virus particles, which is integral for the regulation of survival and proliferation of the infecting HIV-1 and to replication restriction of the invading virus in the host. However, it is unknown whether and how these individual events taking place at different stages of the HIV-1 life cycle are orchestrated as an overall strategy to overcome the restrictions conferred by the host cells. Thus, in this review, we overview the interplay between HIV-1 viral and cellular proteins for restrictions/competitions for proliferation of the virus in the infected cell, which could open a new avenue for the development of therapeutics against HIV-1 via targeting a specific step of the proteasome degradation pathway during the HIV-1 life cycle.

## 1. Introduction

The ubiquitin proteasome system (UPS) is a major intracellular protein degradation system involved in various critical cellular processes in eukaryotes. The UPS consists of the concerted actions of three distinct enzymatic components: E1 ubiquitin-activating enzyme, E2 ubiquitin-conjugating enzyme, and E3 ubiquitin ligase [1,2]. Among these, E3 ubiquitin ligase is the most critical in protein degradation processes, since the enzyme recognizes a target protein to be degraded and transfers ubiquitin (Ub) to the target protein for degradation with specificity [2,3]. The polyubiquitinated proteins will then be recognized by the 19S regulatory particle in an ATP-dependent binding step [4], enter into the 26S proteasome complex, and degrade into amino acids or small peptides, unless the polyubiquitinated proteins are deubiquitinated by the deubiquitinating enzymes (DUBs), such as ubiquitin specific protease (USP) proteins [5]. That is, E3 ubiquitin ligase mediates the last step of the ubiquitination process by recognizing and selecting the target molecules with specificity, and thus most studies have focused on the interaction of the target protein with E3 ubiquitin ligases for understanding the molecular mechanisms of the protein degradation processes by the UPS.

The proteasomal degradation pathway plays a critical role for many cellular processes, including cell cycle [6,7,8], apoptosis [9,10,11], antigen-presentation [12,13,14], etc., and thus aberrations and dysregulations of the UPS can result in the pathogenesis of several important human diseases [15,16,17]. As such, given the general importance of the UPS in a wide variety of cellular processes by regulating protein fate and function, it is reasonable to suggest that HIV-1 and the invaded hosts exploit the UPS in a contest for their survival and proliferation. In fact, several previous reports have indicated that HIV-1 and HIV-1-infected cells exploit cellular machineries for viral and cellular protein decay at every step of the HIV-1 life cycle to restrict their proliferations reciprocally. For instance, Env/Vpu/Nef-mediated degradation of CD4 receptor molecules impedes virus entry and thereby superinfection [18,19,20,21,22,23,24], TRIM5α-triggered Gag is integral to species tropism [25,26,27,28,29,30,31], degradation of tetherin by Vpu is essential for the release of the assembled virus particles [32,33,34,35], and so on. In addition to these interactions between viral and cellular proteins, the interplay between HIV-1 viral proteins is also known to have an integral role in the regulation of the viral replication in the infected cells [36]. A recent publication even indicated that regulation of proteasomal degradation plays a key role in the establishment of viral latency [37]. Thus, we present a review on how the interplay between viral and viral/cellular proteins in their degradation impacts the HIV-1 life cycle, focusing on recent advances in this field of research.

## 2. Proteasomal Degradation in Regulation of the HIV-1 Life Cycle

### 2.1. HIV-1 Life Cycle 

The HIV genome contains at least six regulatory genes, *vif*, *vpr (vpx* for HIV-2), *vpu*, *tat*, *rev*, and *nef*, in addition to three structural genes, *gag*, *pol*, and *env*, which present in all standard retroviruses. Molecular regulation for the stage-specific expression of the viral genes has been studied comprehensively; that is, regulatory genes to produce Tat, Rev, and Nef, are expressed at the early stage, while those for structural proteins, such as Gag, Pol, and Env, are expressed at the late stage of virus infection [38]. Specifically, the Tat expressed at the early stage of the virus infection accelerates the expression of viral genes, and the Rev selectively pumps out the structural genes containing Rev-responsive element (RRE) to the cytoplasm so that the structural proteins are encoded at the late stage of virus infection [38,39,40,41]. Hence, Rev protein plays a key role in the transition from the early to the late stage of the HIV-1 life cycle. Consequently, the HIV-1 life cycle would be dysregulated if the expressed Tat and Rev, including other viral proteins, were not degraded in a timely manner after they completed their duties—the transactivation of the viral gene expressions by Tat and transitioning from the early to the late stage by Rev; that is, elimination processes of these synthesized proteins are quintessential in understanding the regulation of the HIV-1 replication cycle and thereby HIV-1 pathogenesis. This review highlights the impacts of viral and viral/cellular protein degradation on the regulation of the major steps of the HIV-1 life cycle.

### 2.2. Impacts of the Interplay between Viral and Cellular Proteins on the Regulation of the Proteasomal Degradation for the HIV-1 Life Cycle

#### 2.2.1. Virus Entry

According to recent research, the most notable cellular proteins involved in the regulation of HIV-1 entry via the proteasome degradation are SERINC (SERine INCorporator) and IFITM (Interferon-Induced TransMembrane). First, the SERINC family is composed of five members, from SERINC1 to SERINC5, and participate in the transport of serine amino acid through the lipid bilayer and in the biosynthesis of sphingolipids and phosphatidylserine by incorporating serine into membrane lipids [42]. Among the family, SERINC5 and, to a lesser extent, SERINC3 are involved in the impairment of viral infectivity by blocking viral entry [43,44,45], probably through restricting the expansion of the viral fusion pore and thus preventing the release of the viral core into the cytoplasm [43,45]. The SERINC-mediated antiviral activity is counteracted by Nef, wherein Nef induces downregulation of SERINC from the cell membrane by sequestering it in the endosomes for its subsequent degradation by the endo-lysosomal system [43,46,47]. However, the precise molecular mechanisms on how Nef, a virion protein within the virus particle [48,49,50,51], can enhance virus entry are unclear, albeit intracellular Nef from HIV-1-infected cells can ablate SERINC-mediated blockage of the virus entry [43,45,52,53]. With respect to HIV-1 replication, Nef is known to be dispensable or displays slightly, if at all, positive for an in vitro virus replication in the HIV-1 susceptible CD4+ cells [54,55,56], and Nef-mediated counteraction against SERINC is strain dependent (some strains of HIV-1 have Nef-dependent mechanisms to resist high levels of SERINCS, while other strains are independent) [52]. Further, a recent report addressed that the counteraction of SERINC5 by Nef did not increase (or only modestly increased) HIV-1 replication in CD4+ T cells and in ex vivo infected lymphoid tissue or the resting peripheral mononuclear cells (PBMC), respectively [57], casting implications of the significance of the interplay between Nef and SERINC in the enhancement of the virus entry and thus HIV-1 infectivity. In this context, a recent report indicated that HIV-1 envelope (Env), not Nef, is able to resist high levels of SERINC5 without excluding SERINC5 from incorporation into virions [52]. Since HIV-1 infection into its susceptible cells is initiated by the binding of Envelope (Env) protein to the receptor/co-receptor molecules [58,59,60,61], followed by fusion between viral and cytoplasmic membranes by gp41 of Env [62,63], SERINC-mediated inhibition of the correct fusion of the viral Env with the plasma membrane by promoting inactivation of Env [43,45,53], thereby preventing virus entry into host cells, is germane in the explanation of the role of SERINC in HIV-1 entry and thus replication.

IFITMs (IFITM1, 2, and 3) are one of the several interferon (IFN)-stimulated genes (ISGs) that restrict entry of a broad spectrum of enveloped viruses [64,65], including HIV-1 [66,67]. IFITMs-mediated restriction of the virus entry is known to be achieved by blocking the imprinting of the molecules into virions and the processing of Env. However, detailed mechanisms are unclear at this point. Recent reports indicated that IFITMs not only impede HIV-1 entry but also inhibit HIV-1 infectivity in viral producer cells [68,69,70] by negative imprinting of the incorporation of IFITMs into virions [68,69,71] and by impairment of the processing of HIV-1 Env [70]. Even if there were a consensus for the role of IFITMs in the inhibition of HIV-1 entry into the target cells, molecular details of IFITMs-mediated blockage of the virus entry mechanisms remain to be further elucidated. Specifically, it is controversial whether IFITMs distinguish co-receptors in the virus entry, if IFITM-mediated inhibition occurs in a co-receptor-dependent manner, and impair virus entry or viral infectivity of HIV-1 in producer cells [67]. It has been reported that IFITMs impeded HIV-1 differently with different tropism and that IFITMs-mediated inhibition of the virus entry is a co-receptor dependent process, wherein IFITM1 more potently restricts entry of CCR5 viruses, while IFITM2 and 3 are more effective in the blocking of CXCR4 entry [71,72], which is different from other reports [70,73].

Degradation of CD4 by Vpu/Env/Nef is also crucial for virus entry as well as superinfection, which has been well established in previous reports, even if the molecular mechanisms of these viral protein-mediated degradations of CD4 were disparate [18,19,20,21,22,23,24,74]. Since Vpu, Env, and Nef are expressed in a stage-specific manner of the HIV-1 life cycle—that is, Nef is expressed at early stages, while Env and Vpu are encoded from messages harboring Rev-responsive element (RRE) and thus at late stages of the HIV-1 life cycle—intracellular-expressed Nef at the early infection downregulate CD4 by enhancing its internalization and directing the receptor to lysosomes for degradation [20,75,76,77,78,79]. Nef is polyubiquitinated, and diubiquitination of the lysine 144 is required for Nef-mediated CD4 downregulation [80]. In contrast, Vpu and Env reduce the amount of CD4 on the infected cell surface by impairing transport of the newly synthesized CD4 to the cell surface at the late stage of the HIV-1 replication cycle [74,81,82]. Downregulation of the amount of CD4 on the surface of the infected cells by these intracellular viral proteins in HIV-1-infected cells impedes superinfection throughout the HIV-1 life cycle, which is reported to contribute to the prevention of premature cell death and thus to expand the period of effective virus production [21]. Taken together, all of these reports indicate that regulation of the amount of cellular proteins on the surface of the infected cells or of virions by the viral proteins via protein degradation processes plays an integral role in the regulation of virus entry, including superinfection, the initial step of the HIV-1 life cycle, into the host cells.

#### 2.2.2. Intracellular Events

##### Uncoating: TRIM5α/Gag

It has been reported that HIV-1 does not replicate in nonhuman primates [83,84,85], and TRIM5α (TRIpartite Motif-containing protein 5α), a RING-type E3 ubiquitin ligase [86] interacting with the viral capsid, is responsible for the restriction of replication of the entered HIV-1 in a species-specific manner [25,87,88,89]. Specifically, TRIM5α of human or new world monkeys exerts little or no effect on HIV-1 replication, as opposed to that of rhesus (*Macaca mulatta*) or cynomolgus (*Macaca fascicularis*) monkeys [25,84,90,91], showing a key role of TRIM5α in the interspecies barrier for HIV-1 replication [26,27,28,29,30,31]. To elaborate the molecular process of TRIM5α-triggered inhibition of the virus replication, TRIM5α existing as a dimer in the target cell cytoplasm forms a complementary hexameric lattice upon interaction of the TRIM5α SPRY (SPIa and Ryanodine Receptors) domain with the capsid of a restriction-sensitive retrovirus, causing envelopment of the infectious core [92,93,94,95,96]. Then, TRIM5α increases intrinsic E3 ubiquitin ligase activity so that it is autocatalytic, covalently attaching ubiquitin to N-terminal of TRIM5α [95], and if avidity for the viral capsid protein were sufficient, the virion core uncoats prematurely and reverse transcription will be blocked [90,97,98], so that virus replication is ablated. These processes appear to be mediated by the UPS, since treatment with the proteasome inhibitors restores a normal decapsidation and reverse transcription, and TRIM5α and viral cores are co-localized with the proteasome [99,100,101].

Moreover, as noted above, interaction of TRIM5α with the viral capsid enhances its E3-Ub ligase activity and together with the E2 enzyme, UBC13/UVE1A (Ub-conjugating enzyme 13/Ub-conjugating enzyme variant 1A), stimulates TAK1 (Transforming growth factor β-Activated kinase 1), which in turn activates AP1 and NFκB signaling, which are critical for regulation of the expression of inflammatory chemokines and cytokines [94,102]. These reports indicate that in addition to direct antiviral activity, TRIM5α also plays an important role as an innate immune sensor for the retrovirus capsid lattice.

##### Reverse Transcription

(i) APOBEC/Vif

Apolipoprotein B mRNA-editing enzyme, catalytic polypeptide-like 3 (APOBEC3/A3), which induces the transition of cytosine to uracil on single-stranded DNA, is incorporated into budding HIV-1 virions and is thus carried over into the next infected cells [103]. During and after entry, viral genomic RNA is reverse transcribed to generate double-strand DNA, using viral reverse transcriptase, and in the process, APOBEC3G (A3G), prevents the production of functional viral protein by inducing hypermutations on single-stranded DNA, leading to C to U transitions [103,104,105,106,107,108]. These mutations can either be recognized by uracil DNA glycosylases, like the virion-associated UNG2 (Uracil *N*-Glycosylase 2), leading to the degradation of the provirus by abasic site endonucleases [109]. These mutations frequently produce mutated non-functional viral proteins and thereby impair HIV-1 replication.

HIV-1 Vif, a virion protein, essential for efficient virus replication by producing infectious progeny virus particles [110,111,112], is known to counteract cellular restriction factor APOBEC3G by precluding A3G incorporation into virions [105,113,114,115,116,117]. Specifically, binding of Vif to APOBEC3G induces the polyubiquitination of APOBEC3G followed by its proteasomal degradation, preventing its incorporation into HIV-1 virions [118,119]. In the process, Vif interacts with Cullin-RING Ub ligase (CRL) complex (Cullin5, Elongin-B and -C) to generate Vif-BC-Cul5 through a novel SOCS-box motif, and the Vif in the complex associates with APOBEC3G to induce the ubiquitination and degradation of APOBEC3G [115,120,121]. The Vif-BC-Cul5 complex is also known to trigger polyubiquitination followed by proteasomal degradation of another human cytidine deaminase APOBEC3F, another member of the APOBEC3 family, regulating HIV-1 replication [122].

Several previous reports have indicated that Vif-mediated degradation of the host restriction factor APOBEC3 for the inhibition of HIV-1 replication is also counter-regulated. Heat shock protein 70 (HSP70) disrupts interactions between Vif and APOBEC3G and thereby impedes ubiquitination followed by degradation of APOBEC3G [123], and HDAC6 enhances autophagic degradation of Vif to prevent Vif-triggered ubiquitination and degradation of APOBEC3G [124]. The stability of Vif itself is also regulated by various cellular proteins: Mdm2, an E3 Ub ligase, reduces the level of intracellular Vif by inducing ubiquitination and degradation [125,126], and core binding factor β (CBFβ), which is required for the formation of Vif-Cul5 E3 Ub ligase complex, reverses Mdm2-associated degradation of Vif by binding to Vif and thus blocking the interaction of Vif with Mdm2 [127,128,129] Further, proteasomal decay of Vif is also known to be regulated by cyclin F and two HECTs (Homologous to the E6-AP Carboxyl Terminuses)—AIP4 and NEDD [130].

(ii) SAMHD1/Vpr (Vpx)

*Vpx* is present only in HIV-2/SIVsmm/SIVmac, but not in HIV-1 and SIVcpz, while all primate immunodeficiency viruses harbor the *vpr* gene in their genomes [131,132,133]. Since it is believed that *vpx* has evolved from *vpr*, and these genes share many functional similarities, it is believed that lack of *vpx* in HIV-1 can be compensated by the presence of *vpr* [134], which is strongly supported by Gibbs et al., wherein deletion of either gene of SIVmac239 progresses with AIDS to a terminal stage, even if virus burden in the infected rhesus monkeys were lowered and declines in CD4^+^ lymphocytes were delayed, while deletion of both genes severely lowers virus burdens and hence shows no evidence of disease progressions [135]. One of the similarities is that both Vpx and Vpr play a critical role in the ubiquitination followed by degradation of SAMHD1 (Sterile Alpha Motif and Histidine Aspartate domain-containing protein 1), which inhibits reverse transcription by depleting the pool of cellular deoxy Nucleotide Triphosphates (dNTPs) [136,137,138]. For degradation of SAMHD1, Vpx or Vpr recruits the CUL4A-DDB1-DCAF1 (DDB1 and CUL4-associated factor 1) E3 ubiquitin ligase by a direct interaction with its substrate recognition protein, DCAF1 [139], which induces the ubiquitination followed by degradation of SAMHD loaded to the complex by interacting its C-terminal domain with Vpx and Vpr [139,140,141,142]; that is, Vpx and Vpr can enhance diverse primate immunodeficiency virus replication by inducing degradation of the host restriction factor, SAMHD1, and thereby increasing reverse transcription activity.

Recent report showed that Vpx and Vpr also associates with the human silencing hub (HUSH) complex [143,144], FAM208A (TASOR/RAP140), MPHOSPH8 (MPP8), PPHLN1 (PERIPHILIN), and MORC2 [145,146,147], which restricts the replication of primate immunodeficiency viruses [143,144]. It has been reported that Vpx and Vpr counteract the HUSH complex-mediated inhibition of the virus replication by reducing the steady-state level of these proteins in a DCAF1/CUL4A/B/proteasome-dependent manner [148,149], providing explanations of the impacts of Vpx and Vpr on the reporter gene expression, which is not illustrated by SAMHD1 degradation [150,151,152]. Taken together, these reports show that regulation of the stability of Vpx/Vpr and their corresponding cellular partners plays an integral role in the restriction of the infected primate immunodeficiency viruses and the infected cells in regard to their survival and proliferation.

##### Integration of Provirus into Host Chromosome

The reverse transcribed double-strand DNA is transported to the nucleus and incorporated into the host chromosome, wherein viral protein, integrase (IN), plays a main role. Since IN comprises the N-terminal phenylalanine, N-degron signals, the protein is susceptible to rapid degradation by 26S proteasome followed by the class of E3 Ub ligases [153,154]. Substitution of the N-terminal amino acid to methionine increases, but the protein is still short-lived [154,155,156,157,158], indicating that IN is degraded through the proteasomal pathway, independent of N-terminal recognition. In fact, Ali et al. reported that the E3 RING ligase, TRIM33, plays a major role in the determination of the stability of IN by binding to the carboxy-terminal domain of IN through its RING portion and determining its poly-ubiquitination, and lack of TRIM33 rescues the infectivity of HIV-1 [159]. These data indicate that HIV-1 infectivity is impeded by the TRIM33-mediated decrease of IN function by reducing IN stability.

##### HIV-1 Replication (Viral Gene Expression)

(i) Tat

After the establishment of HIV-1 infection by inserting the reverse-transcribed double-strand viral DNA into the host chromosome, HIV-1 Tat increases the steady-state levels of all the viral transcripts by interacting with the TAR (trans-activating responsive) element from the integrated proviral DNA and thereby HIV-1 replication [160,161,162,163,164,165]. Molecular mechanisms on how Tat augments viral gene expression have been thoroughly investigated, while elimination processes of the protein, after it completes its duty, have not been studied comprehensively; that is, regulation of the stability of HIV-1 Tat is key to understanding HIV-1 replication and thus HIV-1-associated pathogenicity. First, it has been reported that HIV-1 Rev causes specific degradation of cytoplasmic Tat and thus inhibits HIV-1 replication [166]. According to the report, Rev induces the degradation of Tat indirectly by down-modulating the expression of NQO1 [NAD(P)H:quinine oxidoreductase 1], which contributes to the stabilization of the Tat protein in a dose-dependent manner, and in the process, the nuclear export signal of Rev is critical [166]. Another report indicated that Nef suppresses the nuclear localization of Tat and thus triggers the degradation of Tat by the proteasome-dependent pathway in the cytoplasm through Hsp70 [36], which is also known to inhibit Vif-mediated ubiquitination and degradation of APOBEC3G [123]. In addition, nucleocapsid (NC) reduces the amount of the intracellular Tat, which is reversed by treatment of MG132 [167]. However, the level of ubiquitination of Tat by NC is unchanged, suggesting that NC-mediated degradation of Tat is ubiquitination-independent [167].

Besides viral proteins, the interplay between Tat and cellular elements involved in proteasomal degradation modulates the replicability of HIV-1. Tat stabilizes Mdm2, an E3 Ub ligase, by activating Akt, which enhances Tat-mediated viral replication [168]. PJA2, a RING finger E3 ligase, polyubiquitinates Tat in a non-degradative manner and regulates the elongation step of transcription of Tat in P-TEFb complex [169]. A recent report showed that a type of long noncoding RNA, called NRON, induces degradation of Tat by directly linking Tat to the Ub proteasome components, including CUL4B and PSMD11, and thus develops latent infection of HIV-1 [170].

(ii) Nef

Activation of the resting CD4^+^ T cells is critical for replication of the infected HIV-1 [171,172,173,174], and Nef is known to activate the cells by stimulating various signaling molecules [175,176,177,178]. That is, proteasomal degradation of key signaling proteins involved in the T cell activation process by Nef alters the replicability of the infected HIV-1 into its susceptible cells. Further, Simmons et al. showed that Nef, in conjunction with c-Cbl, excludes the E2 Ub-conjugating enzyme, UbcH7, from lipid rafts, and suppression of p85Cool-1/βPix in the ternary complex amongst Nef/c-Cbl/p85Coo1/βPix in the lipid rafts attenuates HIV-1 replication [179]. Our unpublished data indicate that UBE3A (Ub protein ligase E3A, also known as E6AP—an E6 associated protein) which causes E6-mediated p53 degradation in HPV (human papilloma virus) interacts with Nef and inhibits HIV-1 replication. Further, a previous report also showed that the addition of MG132 or lactacystin, each a specific inhibitor of cellular proteasome activity, preferentially increases cellular permissiveness of *nef*-deficient rather than wild type HIV-1 [180]. Taken together, all these reports denote the significance of the proteasomal degradation in the regulation of HIV-1 replication.

#### 2.2.3. Release of Virus Particles—Budding:

(i) Tetherin/Vpu

At the last stage of the HIV-1 life cycle, the newly assembled virions remain tethered to the plasma membrane without being released, and are eventually endocytosed and degraded in the absence of Vpu [181,182], demonstrating that Vpu is required to release the assembled virus particles to the outside of the infected cells. Virion-tethering to the plasma membrane is mediated by two domains—a transmembrane domain proximal to the N-terminus and an extracellular C-terminal glycosyl-phosphatidylinositol anchor domain—of tetherin/BST-2 [182], and interaction of Vpu with tetherin is known to sequester BST-2 away from the virion budding sites in intracellular compartments, particularly the trans-Golgi-network, thereby allowing the release of the assembled virions [182,183,184,185,186]. In the process, Vpu recruits E3 ubiquitin ligase adaptor, β-TrCP, by the cytoplasmic DSGxxS motif [187], which may lead to ubiquitination of tetherin by the cellular E3 ubiquitin ligases, March8 and NEDD4 (Neural precursor cell Expressed Developmentally Down-regulated protein 4) [188] followed by degradation in the endo-lysosomal system [185,189].

Vpu-triggered downregulation of tetherin on the surface of the HIV-1-infected cells is also known to be achieved without involvement of β-TrCP [185,190,191]. Vpu binds and targets tetherin to ESCRT (Endosomal Sorting Complex Required for Transporter)-dependent endosomal degradation via interaction with clathrin adaptor AP-1 and -2, wherein phosphorylation of Vpu is required [192,193,194]. Reduction of the surface expression of tetherin by Vpu is also exploited by the autophagy pathway. Vpu induces the removal of tetherin from HIV-1 budding sites by interacting with ATG8 (Autophagy related protein 8), ortholog LC3, thereby leading to phagocytosis [195]. However, Vpu-associated degradation of tetherin might not be an absolute requirement for budding of the assembled virions, since Vpu is capable of intracellular sequestration of tetherin without degradation [34,186].

When Vpu is deficient, as is observed with SIVagm, SIVblu, and SIVmac, Nef takes over the role of tetherin counteraction [32,183,184], and in HIV-2, Env substitutes for Vpu function in releasing virions [196,197]. Thus, efficient virus budding in the *vpu*-deficient primate immunodeficiency viruses is implemented by other viral genes.

(ii) ESCRT/NEDD/Gag

Two late (L) domains, the PTAP and YPXnL, (where X refers to any amino acid and *n* = 1 to 3 residues) identified within the p6 region of the HIV-1 Gag play an essential role in the budding of the assembled virus, which is confirmed by the mutational analyses in the motifs [198,199]. The PTAP motif binds to the cellular protein Tsg101 [200,201,202], which functions in HIV-1 budding [200,203,204], while LYPXnL recruits Alix/AIP-1 [205,206]. Over-expression of a fragment containing Bro1 with the adjacent V domain of Alix inhibits release via both the PTAP/Tsg101 and the LYPXnL/Alix pathways in a nucleocapsid (NC)-dependent manner, and provision of a link to the ESCRT machinery through NEDD4-2s over-expression rescues NC mutant-associated release defects [207,208], indicating that the NC region of HIV-1 Gag cooperates with the two L domains to recruit the cellular machinery necessary for viral budding [209] and NEDD-like ligase is critical to function in virus release [207,208,209], illustrating the significance of the ESCRT/NEDD pathway in the release of the virus.

## 3. Concluding Remarks and Future Directions

The above assessments reveal that the proteasomal degradation plays an integral role in regulating the degradation of viral and cellular proteins at every step of the HIV-1 life cycle, which is vital for the survival and proliferation of the virus and the HIV-1-infected host cells. However, even if the molecular processes of the individual degradation event in the HIV-1 life cycle were well elucidated, as described above and Figure 1, the significance of the degradation of certain viral and/or cellular proteins with respect to the virus replication is still elusive. One of the reasons is the dispensability of certain viral genes in the virus replication. For instance, *vpr*, *vpx*, or *nef* are basically dispensable for efficient virus replications in the T cell line [54,55,135,210,211], and thus the impact of the proteasomal decay of these proteins encoded from these genes on the replicability of the primate immunodeficiency viruses is obscure. Another important point is that multiple viral proteins are involved in the degradation of the identical target molecules, and thus the specific contribution of each viral protein to the virus replication by proteasomal degradation of the target protein could be murky. For instance, both Env and Nef affect SERINC-associated virus entry, Vpu/Env/Nef regulate the down-modulation of CD4 on the HIV-1-infected cell surface, and both Vpx and Vpr in HIV-2 and certain SIVs alter reverse transcription activity by interplaying with SAMDH, as described above. Finally, there is a need for the elucidation of the molecular mechanism(s) regarding how these viral proteins that localize different subcellular organelles in the infected cells and are expressed at different stages of HIV-1 replication cycle (early vs. late) coordinate or orchestrate protein degradation. It is also needed to answer questions about how the identical cellular partners regulate or are regulated by the same viral element. Elucidation of how these particular events are coordinated among different virus stages, and of which molecules play key roles in viral and cellular protein fate determination, will fortify our understanding of fine-scale life cycle stage transitions and proteasomal degradation-mediated HIV-1 pathogenicity, fostering antiviral therapeutics targeting a specific step in proteasome degradation processes.

## Figures and Tables

**Figure 1 ijms-20-02984-f001:**
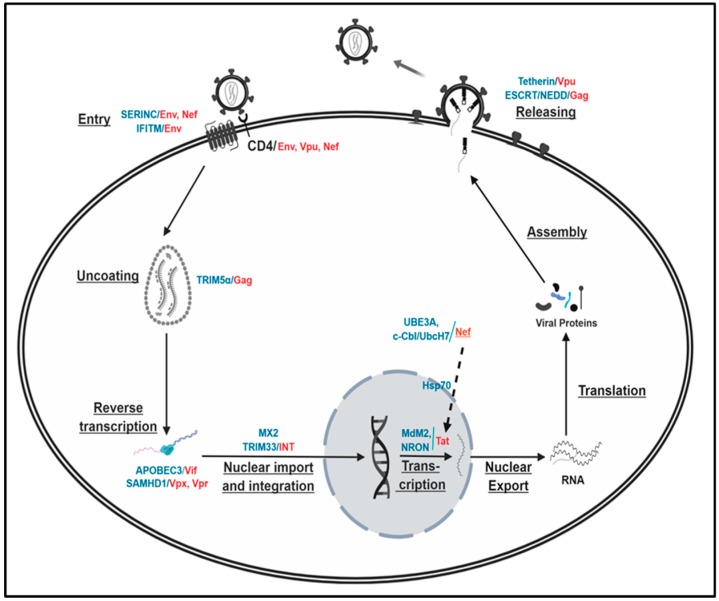
Schematic representation of the HIV-1 life cycle. Each step of the HIV-1 life cycle is underlined, and the viral and cellular proteins involved in each step of the proteasomal regulation are shown as red and teal, respectively. Proteasomal regulation between viral proteins is depicted as a dotted line. See the text for the detailed description of the role of the indicated viral and cellular proteins for the proteasomal degradation processes.

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
