# Peer review of "Role of the Ubiquitin Proteasome System (UPS) in the HIV-1 Life Cycle"

_ijms, 2019, doi:10.3390/ijms20122984_

Reviewer 1 Report

Authors reviewed literature on proteasomal degradation in HIV life cycle. This is an interesting and relevant topic, which has wider implications, not only in understanding the HIV biology but also in developing novel therapeutics.

They take an interesting approach in structuring the text from viral replication/viral protein’ perspective. Authors can also consider including a standalone section on pharmacological targeting of these pathways in the context of HIV infection together with its appropriateness/state of art on relevance for drug discovery. They can also include a toolbox (in form of a table) of reagents that can used to perturb proteasomal machinery.

The writing style comes across as a bit complex sometimes and there’s a need to simplify the text at some instances. Very long sentences should generally be avoided. For example, Page 1 / Line 10: The very first sentence of abstract is too long and not very clear, needs to be simplified.

Minor points:

Page 1 / Line 10: In the first line of abstract: is it were or where?

Line 55: What is TRIM5a-triggered Gag?

Line 100: ref 52 appears to be an incorrect reference if authors are making a point about strain specific effects of nef on SERINCS

How about Heigele et al 2016 CHM

Line 227: “Further, Re previous….” Sentence seem incomplete

Author Response

The writing style comes across as a bit complex sometimes and there’s a need to simplify the text at some instances. Very long sentences should generally be avoided. For example, Page 1 / Line 10: The very first sentence of abstract is too long and not very clear, needs to be simplified.

Minor points:

Page 1 / Line 10: In the first line of abstract: is it were or where?

 [Response] To clarify the sentence, we added "that" in between "Given" and "the" in the first sentence, and it's "were", not "where", in the sentence, since all verb "be" should be "were", when the sentence is led by "if, given that, granted that, etc".

Line 55: What is TRIM5a-triggered Gag?

 [Response] It is now changed to "TRIM5a-triggered degradation of Gag".

Line 100: ref 52 appears to be an incorrect reference if authors are making a point about strain specific effects of nef on SERINCS

How about Heigele et al 2016 CHM 

[Response] We revisited the reference and confirmed that the reference describes differential role of Env and Nef on SERINC-mediated inhibition of virus entry.

Line 227: “Further, Re previous….” Sentence seem incomplete

[Response] We revised the sentence by removing "Rc".

Reviewer 2 Report

Authors of the review titled ‘Role of the Ubiquitin Proteasome System (UPS) in the HIV-1 Life Cycle’ have made an attempt in putting together the literature on the cellular or viral factors that utilize the UPS for their survival. Although, there are reviews existing on the similar topics with respect to the HIV-1, and UPS, authors have tried to explain the same phenomena in the context of viral life cycle and they were successful to some extent. The review requires further improvement to be considered for publication. I also think having a pictorial depiction of some sections in the review instead of just one conclusive figure, would be very helpful for the readers. Please find some of the comments below:

Comments:

1)    At many places in the review, the sentences are quite long and it becomes very difficult for reader to follow the information, and/or arguments authors want to put forward. The review has to be thoroughly checked for such long sentences and make them short, and succinct.

2)    The introduction section does not clearly describe the UPS. It can be obvious for the experts in the field of UPS but for others it can be very confusing to follow. For example, it is better to start with, what is the general function of UPS and then jump into the details of E1, E2, and E3 ligases.

3)    In introduction and section 2.1: HIV is not introduced well and HIV-1 life cycle is not properly explained. A brief overview might help reader to easily follow the next sections of the review. AIso, same sentences (line#75-76), as in abstract, were used. Not sure if one can do that?

4)    Section 2.2.1: lines# 88-90 are same as lines#95-97? Authors talk about role of IFITMs in the viral entry restrictions. But authors fail to mention the possible or probable role of ubiquitination or UPS, if any, in the same.

5)    Lines#212-215, line#244-245, and lines#277-280 are not clear. Please rephrase.

6)    Section on Nef (page#6): the paragraph needs a more specific conclusion than just general one.

7)    Section-Conclusions: I do not fully agree with authors’ final sentence of ‘therapeutics targeting a specific step in proteasome degradation’ as an anti-viral therapeutics development strategy. This contradicts the purpose of the review where authors explain that there are complex network of events/factors from host, here UPS, or virus, in HIV-1 life cycle. Targeting a single step will not serve the purpose. Please comment?

Minor formatting required at place: The abbreviations appear at place without any previous mention about the same. At places it is written an ‘diubiquitination’ instead of ‘deubiquitination’  

Author Response

Comments:

1)    At many places in the review, the sentences are quite long and it becomes very difficult for reader to follow the information, and/or arguments authors want to put forward. The review has to be thoroughly checked for such long sentences and make them short, and succinct.

[Response] We agree with the reviewers' points and therefore rephrased those to be more comprehensive sentences.  For instances, we added "that" in between "Given" and "the" in the first sentence in "Abstract" and removed seemingly redundant phrase in the first sentence on page 3, etc.

2)    The introduction section does not clearly describe the UPS. It can be obvious for the experts in the field of UPS but for others it can be very confusing to follow. For example, it is better to start with, what is the general function of UPS and then jump into the details of E1, E2, and E3 ligases.

[Response] As the reviewer suggested, we added one sentence describing the UPS.

3)    In introduction and section 2.1: HIV is not introduced well and HIV-1 life cycle is not properly explained. A brief overview might help reader to easily follow the next sections of the review. AIso, same sentences (line#75-76), as in abstract, were used. Not sure if one can do that?

[Response] In response to the reviewer's point, we introduced one sentence in HIV-1 life cycle to explain HIV-1 genome.  The sentences were rephrased to avoid repetition of the same sentences.

4)    Section 2.2.1: lines# 88-90 are same as lines#95-97? Authors talk about role of IFITMs in the viral entry restrictions. But authors fail to mention the possible or probable role of ubiquitination or UPS, if any, in the same.

[Response] The sentences were modified to avoid reiteration.   IFITMs-mediated restriction of the virus entry is known to be achieved by blocking imprinting of the molecules into virions and processing of Env.  However, detailed mechanisms are unclear at this point, which is stated in the manuscript.

5)    Lines#212-215, line#244-245, and lines#277-280 are not clear. Please rephrase.

[Response]  According to the reviewer's request, all sentences have been rephrased.

6)    Section on Nef (page#6): the paragraph needs a more specific conclusion than just general one.

[Response] We amended the paragraph to reach a more specific conclusion.  

7)    Section-Conclusions: I do not fully agree with authors’ final sentence of ‘therapeutics targeting a specific step in proteasome degradation’ as an anti-viral therapeutics development strategy. This contradicts the purpose of the review where authors explain that there are complex network of events/factors from host, here UPS, or virus, in HIV-1 life cycle. Targeting a single step will not serve the purpose. Please comment?

Minor formatting required at place: The abbreviations appear at place without any previous mention about the same. At places it is written an ‘diubiquitination’ instead of ‘deubiquitination’

[Response] At one place, we employed "diubiquitination" to explain "di-ubiquitination", while at other places, we used "deubiquitination" to describe removal of Ub from substrate.  At places, we did not spell-out, since abbreviation of those molecules, such as CD4, is already very well established.  However, we will amend with the reviewer's request.